# RECOMP: IMPROVING RETRIEVAL-AUGMENTED LMS WITH COMPRESSION AND SELECTIVE AUGMENTATION

**Fangyuan Xu[1], Weijia Shi[2], Eunsol Choi[1]**
Department of Computer Science
[1]The University of Texas at Austin, [2]University of Washington
{fangyuan,eunsol}@utexas.edu, swj0419@cs.washington.edu

## ABSTRACT

Retrieving documents and prepending them in-context at inference time improves performance of language model (LMs) on a wide range of tasks. However, these documents, often spanning hundreds of words, make inference substantially more expensive. We propose compressing the retrieved documents into textual summaries prior to in-context integration. This not only reduces the computational costs but also relieves the burden of LMs to identify relevant information in long retrieved documents. We present two compressors – an *extractive compressor* which selects useful sentences from retrieved documents and an *abstractive compressor* which generates summaries by synthesizing information from multiple documents. Both compressors are trained to improve LMs' performance on end tasks when the generated summaries are prepended to the LMs' input, while keeping the summary concise. If the retrieved documents are irrelevant to the input or offer no additional information to LM, our compressor can return an empty string, implementing selective augmentation. We evaluate our approach on language modeling task and open domain question answering task. We achieve a compression rate of as low as 6% with minimal loss in performance for both tasks, significantly outperforming the off-the-shelf summarization models. We show that our compressors trained for one LM can transfer to other LMs on the language modeling task and provide summaries largely faithful to the retrieved documents.[1]

## 1 INTRODUCTION

Retrieval-augmented language models (RALMs) (Khandelwal et al., 2019; Izacard et al., 2022; Lewis et al., 2020; Borgeaud et al., 2022) have shown impressive performance on knowledge-intensive tasks (Kwiatkowski et al., 2019; Petroni et al., 2021). Simply prepending retrieved documents to the input without updating the language models (LMs) (Shi et al., 2023b; Ram et al., 2023; Si et al., 2022) allows retrieval augmentation even for black-box LMs, but such approach comes with limitations. First, it increases computational costs as LMs now encode substantially more tokens. Second, even if we manage to adapt LMs to incorporate long context (Beltagy et al., 2020; Zaheer et al., 2020), these models struggle to use all information in the context, frequently missing information placed in the middle (Liu et al., 2023). Third, prepending multiple documents in-context can further *confuse* LMs with irrelevant information, degrading model performances (Mallen et al., 2022; Shi et al., 2023a).

To overcome such limitations, we propose **RECOMP** (**Re**trieve, **Com**press, **P**repend), an intermediate step for RALMs which compresses retrieved documents into a textual summary prior to in-context augmentation. Figure 1 illustrates our approach. The generated summary should be concise to maximize efficiency, be faithful to the retrieved evidence documents, and guide RALM to generate desired outputs when prepended to the input. We train compressors with a learning objective that encourages both efficiency and effectiveness for a target LM. Our framework enables selective augmentation, by generating an empty summary when the retrieved documents are irrelevant or unhelpful for target task and the target LM.

We propose two kinds of compressors: (1) *Extractive compressor* which selects relevant sentences from retrieved document set; (2) *Abstractive compressor* which generates a summary synthesizing

---

[1]Our code is available at https://github.com/carriex/recomp.

Figure 1: An illustration of **RECOMP**, which compresses retrieved documents into a texual summary before prepending it as input to a language model at inference time. The compressed summary guides the LM to generate the correct answer, while reducing the computation costs required to encode the documents. We describe training procedures for the compressors in Section 3.

information from multiple retrieved documents. Both compressors implement multi-document query-focused summarization (Xu & Lapata, 2020), where we summarize retrieved evidence document set with respect to the input query. As we aim to enable RALM to generate correct output when summary is prepended to the input query, we design training schemes to optimize the end task performance. Our extractive compressor is trained with a contrastive learning objective to identify sentences that lead to target outputs, and our abstractive compressor is distilled (West et al., 2022) from an extreme-scale LM (e.g. GPT-3), which achieves impressive summarization performance.

Our experiments show that **RECOMP** can improve performance of frozen LMs on language modeling (Merity et al., 2016) and three question answering datasets (Natural Questions (Kwiatkowski et al., 2019), TriviaQA (Joshi et al., 2017) and HotpotQA (Yang et al., 2018)), while prepending significantly fewer tokens compared to RALM without compression. We present two oracle compression methods – an extractive oracle which selects a sentence in evidence documents that leads to the best task performance and an abstractive oracle which chooses between a summary generated by extreme-scale LLM (e.g. GPT-3) and no retrieval augmentation that leads to the best task performance. Both oracle methods achieve a compression rate as low as 6% and significantly *outperforms* prepending full documents. Our trained compressors also show promising results. For language modelling, both trained compressors achieve a compression ratio of 25% with minimal performance drop. When applied to QA datasets, our best model compresses the documents to 5 - 10% of the original tokens with at most less than 10% relative performance drop and 2x inference time speed up. We conclude with careful analyses of our approach that reveal both its strength and weaknesses, thereby building foundation for future work.

## 2 PROBLEM FORMULATION: **RECOMP**

Given an input sequence $\mathbf{x}$, a target output sequence $\mathbf{y}$ and a set of $N$ retrieved documents $D$ ($[d_1, d_2, ...d_N]$),[2] **RECOMP** compresses retrieved documents $D$ with respect to $\mathbf{x}$ into a summary $\mathbf{s}$ which captures core information in $D$ relevant to $\mathbf{x}$ with significantly fewer tokens than $D$. Our architecture consists of two modules: compressor $c_\theta$ and LM $M$. In this work, we assume a blackbox LM and train the compressor. Given the set of retrieved $N$ documents ($[d_1, d_2, ...d_N]$) and the input sequence $\mathbf{x}$, a compressor returns a token sequence $\mathbf{s}$. We design our compressor to be substantially smaller than LM $M$, as we aim to reduce computational costs of encoding a set of retrieved documents.

The output from compressor, $\mathbf{s}$, should be: (1) **Concise**: The summary should be as short as possible to optimize efficiency. If the retrieved documents do not contain relevant information or retrieval augmentation is not necessary, $\mathbf{s}$ can be an empty sequence. (2) **Effecive**: when $\mathbf{s}$ is prepended to input sequence $\mathbf{x}$ and provided to LM $M$ as a prompt, LM should generate the target output sequence $\mathbf{y}$. (3) **Faithful**: $\mathbf{s}$ should be a faithful and interpretable summary of the input document set (i.e., $\mathbf{s}$ must be entailed by the input document set ($[d_1, d_2, ...d_N]$)). We focus on training compressors for conciseness and effectiveness. We summarize the key ideas for our two compressors, extractive compressors and abstractive compressor here, and discuss their training schemes formally in Section 3.

**Extractive Compressor**  Given $n$ sentences $[\mathbf{s_1}, \mathbf{s_2}...\mathbf{s_n}]$ in the input document set ($[d_1, d_2, ...d_N]$), we train a dual encoder model $\text{enc}_\theta$ which embeds sentence $\mathbf{s_i}$ and the input sequence $\mathbf{x}$ into fixed-

---

[2]Improving retriever is not the focus of this work, so we assume a set of retrieved documents are provided.

dimensional embeddings respectively. Their inner product represents how helpful it would be for the LM $M$ to prepend $s_i$ to the input $x$ to generate $y$. The final summary $s$ from the compressor will be a concatenation of top N sentences ranked by their inner product with the input. As this approach is extractive, we assume the faithfulness criteria is mostly satisfied.[3]

**Abstractive Compressor**  We train an encoder-decoder model `encdec`$_\theta$ to serve as an abstractive compressor, which takes the input sequence $x$ and a concatenation of retrieved document set $D$ $[d_1; d_2; ...d_N])$ and output a summary $s$. Although we do not have human annotations to train this model, prior work (Goyal et al., 2022; Chen et al., 2023; Potluri et al., 2023) suggests that the extreme-scale LMs can generate good query-focused summaries when prompted carefully. Yet, using an extreme-scale model as the compressor is not desirable as we want the compressor to be substantially smaller than the LMs. Thus, we perform **distillation** (Hinton et al., 2015) of extreme-scale LMs to build a lightweight abstractive compressor `encdec`$_\theta$. We do not train specifically for faithfulness, but later manually evaluate the faithfulness in Section 6.

## 3 LEARNING THE COMPRESSORS

Our compressor resembles text summarization models in that the output should be faithful to the original input, yet the main goal is different. Instead of capturing salient information for humans readers, compressors aim to produce a concise text that are useful for a LM on an end task. In this section, we describe how to train the extractive compressor (§3.1) and the abstractive compressor (§3.2) leveraging end task signals. Further training details can be found in the Appendix A.2.

### 3.1 EXTRACTIVE COMPRESSION

As we formulate extractive compression as a ranking problem, training extractive compressor resembles training a reranker for the retrieved documents[4] with two differences. First, our compressor considers a different granularity of input (sentence) compared to the initial retrieval unit (paragraph). Second, the sentence is evaluated based on whether it is useful as input for the LM $M$ on the downstream task (Shi et al., 2023b; Ram et al., 2023).

**Model**  We train a dual-encoder model `enc`$_\theta$ which encodes the input context $x$ and the candidate sentence $s_i$ separately. We obtain an embedding of $x$ and $s_i$ by taking the representation of the `[CLS]` token respectively, and compute their similarity by their inner product. We initialize our model with the contriever checkpoint (Izacard et al., 2021). This model consists of 110M parameters, satisfying the efficiency desideratum of compressor.

**Training**  Figure 2 presents pseudocode for training an extractive compressor with contrastive loss for the language modeling task. For each input query $x_i$, we identify positive and negative sentences from retrieved documents.

For each pair of input sequence $x_i$ and candidate sentences $s_j$, we measure $\textbf{Score}(M, y_i, [s_j; x_i])$ $=$ $\log p_M(y \mid [s_j; x_i])$, log likelihood assigned to target output according to LM $M$ when candidate

---

**Input:** Base LM $M$, Compressor `enc`$_\theta$, Training data $\{x_i, S_i, y_i\}_1^T$ where $x_i$ is input, $S_i = \{s_j\}_1^n$ is a set of candidate sentences from the retrieved documents for $x_i$, $y_i$ is the target answer, and score threshold $\epsilon$.
**Output:** An updated extractive compressor encoder `enc`$_\theta$

1: $\mathcal{T} \leftarrow \emptyset$
2: **for** $i \in \{1, \ldots, T\}$ **do**
3:     $p_i \leftarrow \arg\text{Max}_{s_j \in \{S_i\}} \textbf{Score}(M, y_i, [s_j; x_i])$
4:     **for** $j \in \{1, \ldots, n\}$ **do**
5:         $\mathcal{L} \leftarrow \emptyset$
6:         **if** $\textbf{Score}(M, y_i, [s_j; x_i]) + \epsilon < \textbf{Score}(M, y_i, [p_i; x_i])$ **then**
7:             $\mathcal{L} \leftarrow \mathcal{L} \cup s_j$
8:     **if** $|\mathcal{L}| > 0$ **then**
9:         $\mathcal{N}_i \leftarrow \arg\text{Top5}_{s_j \in \mathcal{L}}(\langle \text{enc}_\theta(s_j), \text{enc}_\theta(x_i) \rangle)$
10:        $\mathcal{T} \leftarrow \mathcal{T} \cup \{(x_i, p_i, \mathcal{N}_i)\}$
11: `enc`$_\theta$ = **Finetune**(`enc`$_\theta$, $\mathcal{T}$)

Figure 2: Learning an extractive compressor for language modeling task.

---

[3]Recent work (Zhang et al., 2022) shows that extractive approach does not always preserve faithfulness, but such cases are still rare compared to abstractive approaches which can easily hallucinate.

[4]Ram et al. (2023) proposes a document reranker based on a cross-encoder model, which is a similar set-up to our sentence selector, but less compute efficient.

sentence is prepended to the input. We consider the sentence with the highest log likelihood as a positive example $\mathbf{p_i}$ (line 3). To construct negative examples $\mathcal{N}_i = \{n_k\}_{k=1}^5$, we choose up to five sentences with top contriever score that has the log likelihood lower than the positive sentence for a threshold(line 6).

Training a compressor for QA task works similarly, but scoring will evaluate whether the LM will generate the correct answer with summary prepended (change in line 6). Pseudo code for the QA tasks is in Figure 6 the Appendix. We train our encoder with a contrastive loss (Karpukhin et al., 2020a), maximizing the similarity between positive pairs $(\mathbf{x_i}, p_i)$ and minimize negative pairs $(\mathbf{x_i}, N_i)$. The training objective is to minimize $-log \frac{e^{sim(\mathbf{x_i}, p_i)}}{e^{sim(\mathbf{x_i}, p_i)} + \sum_{n_j \in N_i} e^{sim(\mathbf{x_i}, n_j)}}$.

**Data**  For the language modeling task, we generate training data using the training split of the Wikitext-103 dataset, selecting the top 20 sentences from the top 5 BM25 retrieved documents for each input context $\mathbf{x}$. For the QA tasks, we generate training data using the training split and consider the top 20 sentences from the top 5 contriever-ms-marco [5] retrieved documents. We report detailed statistics for the training data in Table 5 in the appendix. For each sentence from the retrieved documents, we prepend the Wikipedia page title to it to for decontextualization.

## 3.2  ABSTRACTIVE COMPRESSION

We distill the query-focused summarization ability of extreme-scale LM by generating training dataset with it, filtering the generated data, and training an encoder-decoder model from the filtered dataset (West et al., 2022). In contrast to prior work (Jung et al., 2023) which use intrinsic summarization metric for filtering, we filter with the LM's end task performance with the generated summaries prepended. Fig. 3 presents pseudo algorithm for abstractive compressor training.

### 3.2.1  CREATING TRAINING DATASET FOR DISTILLATION

**Generation From Teacher Model**  For the language modeling task, we manually construct four prompts to summarize evidence document set ($\{\mathbf{p_i}\}_1^n$).[6] Given an input $\mathbf{x_i}$, a retrieved document set $\mathbf{D}_i$, and a prompt $\mathbf{p_j}$ to summarize the document set with respect to the input, GPT-3.5 [7] generates a summary (line 3).

**Filtering with Critic**  After generating a summary for each prompt template, we select the summary which results in the highest end task performance for each example ($\mathbf{s_t}$) as the target summary (line 4-8). **Score**$(M, \mathbf{y_i}, [\mathbf{s_j}; \mathbf{x_i}])$ is the same as the extractive compressor above. We then compare the end task performance with the target summary prepended and with input $\mathbf{x_i}$ only (i.e. no retrieval) on base model $M$ (line 6). If the end task performance gets worse (e.g., increase in perplexity) when prepending the summary, we set the target summary to an empty string (line 7), otherwise we add the target summary to the training set (line 9). This allows for selective augmentation and mitigates the risk of prepending irrelevant documents.

**Input:** Teacher LM $M_t$, LM $M$, Summarization prompt set $\{\mathbf{p_i}\}_1^n$, Compressor encdec$_\theta$, Training data $\{\mathbf{x_i}, \mathbf{D_i}, \mathbf{y_i}\}_1^T$ where $\mathbf{x_i}$ is input, $\mathbf{D_i}$ is the set of retrieved document for $\mathbf{x_i}$, $\mathbf{y_i}$ is the target answer.
**Output:** An updated encdec$_\theta$

1: $\mathcal{T} \leftarrow \emptyset$
2: **for** $i \in \{1, \ldots, T\}$ **do**
3:    $v_r \leftarrow -\infty$
4:    **for** $j \in \{1, \ldots, n\}$ **do**
5:       $\mathbf{s_j} = \mathbf{Decode}(M_t, [\mathbf{p_j}; \mathbf{x_i}; \mathbf{D_i}])$
6:       $v_j = \mathbf{Score}(M, \mathbf{y_i}, [\mathbf{s_j}; \mathbf{x_i}])$
7:       **if** $v_j > v_r$ **then**
8:          $\mathbf{s_t} \leftarrow \mathbf{s_j}, v_r \leftarrow v_j$
9:    $v_d = \mathbf{Score}(M, \mathbf{y_i}, [\mathbf{x_i}])$
10:    **if** $v_r < v_d$ **then**
11:       $T \leftarrow T \cup \{(\mathbf{x_i}, \mathbf{D_i}, \emptyset)\}$
12:       continue
13:    $T \leftarrow T \cup \{(\mathbf{x_i}, \mathbf{D_i}, \mathbf{s_t})\}$
14: encdec$_\theta$ = $\mathbf{Finetune}($encdec$_\theta, T)$

Figure 3: Learning an abstractive compressor for language modeling task.

Constructing training datasets for the question answering tasks works similarly, with the following modifications. As summarization for the question answering task is more straightforward, we use a

---

[5] https://huggingface.co/facebook/contriever-msmarco
[6] The exact prompts can be found in Table 14 in A.1.
[7] We use gpt-3.5-turbo in all our experiments.

single prompt for each dataset. We filter out examples where prepending the summary does not lead to performance improvement. Pseudo code for the QA tasks is in Figure 7 in the Appendix.

**Model & Training**    We use encoder-decoder LM (775M), initialized from T5-large checkpoint (Raffel et al., 2020). This model has been trained with summarization datasets (Hermann et al., 2015).

**Data**    We summarize top 5 retrieved documents for both language modeling and question answering tasks. We generate training examples using subsets of the training set for the Wikitext-103 dataset, the entire NQ training set and TriviaQA training set. For HotpotQA, we only generate abstractive summaries for the subset of training data (56%) where the gold answer is in the retrieved documents to reduce API costs. We report percentage of data filtered and empty summaries in Table 5 in A.1.

## 4    EXPERIMENTAL SETTINGS

We evaluate our approach on language modeling and open-domain QA following prior work (Shi et al., 2023b; Ram et al., 2023). For both tasks, we report the task performance as a measure of effectiveness and the number of tokens provided in context as a measure of efficiency.

### 4.1    LANGUAGE MODELING

We evaluate language modeling perplexity on WikiText-103 (Merity et al., 2016) benchmark on three open-sourced LMs of varying scale: GPT2 (117M), GPT2-XL (1.5B; Radford et al. (2019)) and GPT-J (6B; Wang & Komatsuzaki (2021)). We train our compressors using GPT2 as the base model and evaluate whether the trained compressor transfer to GPT2-XL and GPT-J. We use the BM25 retriever (Robertson & Zaragoza, 2009) to retrieve from the Wikipedia corpus from Dec. 20, 2018 (Karpukhin et al., 2020a). The articles are truncated into non-overlapping documents of 100 words. During retrieval, articles containing the input sequence $\mathbf{x}$ is removed from the corpus to prevent data contamination. Following Ram et al. (2023), we perform retrieval every 32 tokens.

### 4.2    OPEN-DOMAIN QA

**Datasets**    We evaluate our model on three benchmark dataset: Natural Questions (NQ) (Kwiatkowski et al., 2019), TriviaQA (Joshi et al., 2017)) and HotpotQA (Yang et al., 2018). We report results on development set of NQ, test set of TriviaQA and randomly sampled 500 examples from HotpotQA development set. We report Exact Match (EM) and token-level F1 of answer strings.

**Base Language Models & Retrieval Corpus**    We use Flan-UL2 (20B)(Chung et al., 2022), a large scale instruction-tuned LM. We use contriever model trained on MS MARCO dataset (Campos et al., 2016) as a retriever on Wikipedia corpus from Dec. 20, 2018 for all three datasets. The articles are truncated into non-overlapping documents of 100 words.

**Prompt Format**    We include few-shot in-context examples in the prompt, followed by the retrieved documents and the question. We use five randomly sampled in-context examples from training set, which constitutes 110, 147, and 149 tokens on average for NQ, TQA and HotpotQA respectively. We concatenate retrieved documents in ascending order of retrieval score, with the highest scored one closest to the question (Si et al., 2022). We do not include the retrieved documents for in-context examples as it did not improve performance. An example input can be found in Appendix Table 13.

### 4.3    BASELINES AND ORACLES

**Baselines**    We first consider two heuristic token and phrase-level compression methods: **BoW**, which converts the retrieved documents to a list of ordered unigram and concatenates them together and **Named Entities (NE)**, which extracts a list of ordered named entities from retrieved documents and concatenates them. For the extractive compressor on the language modeling task, we use **BM25** and **Contriever** Izacard et al. (2021), which rank the sentences by their similarity to the input $\mathbf{x}$ as baselines. For the QA datasets, we report results using **BM25**, **Contriever** finetuned on MS MARCO and **DPR** (Karpukhin et al., 2020b) fine-tuned on NQ. We also report a **Random** baseline which

Table 1: Results on language modeling task. We report results on GPT-2, GPT2-XL and GPT-J with compressors trained with GPT-2.

| | In-Domain | | Out-Domain | | | |
| | GPT2 (117M) | | *GPT2-XL (1.5B)* | | *GPT-J (6B)* | |
| **In-context Evidence** | **# tokens** | **PPL** | **# tokens** | **PPL** | **# tokens** | **PPL** |
|---|---|---|---|---|---|---|
| - | 0 | 37.84 | 0 | 19.89 | 0 | 11.44 |
| *RALM without compression* | | | | | | |
| Top 1 document | 141 | 32.90 | 141 | 17.86 | 141 | 10.57 |
| Top 5 documents | 512 | 35.53 | - | - | - | - |
| *Phrase/token level compression* | | | | | | |
| Top 1 document (BoW) | 66 | 36.13 | 66 | 18.85 | 66 | 10.97 |
| Top 1 document (NE) | 34 | 37.23 | 33 | 19.67 | 33 | 11.39 |
| *Extractive compression of Top 5 documents (select top 1 sentence)* | | | | | | |
| *Oracle* | 32 | 30.36 | 32 | 16.58 | 31 | 9.92 |
| *Oracle (w/ gpt2)* | 32 | 30.36 | 32 | 16.99 | 32 | 10.22 |
| Random | 27 | 36.98 | 27 | 19.55 | 27 | 11.32 |
| BM25 | 33 | 36.63 | 33 | 19.02 | 33 | 11.08 |
| Contriever | 33 | 35.54 | 33 | 18.98 | 33 | 11.05 |
| Ours (init. w/ Contriever) | 31 | 33.67 | 31 | 18.19 | 31 | 10.73 |
| *Abstractive compression of Top 5 documents* | | | | | | |
| *Oracle* | 68 | 30.67 | 66 | 16.87 | 65 | 10.10 |
| *Oracle (w/ gpt2)* | 68 | 30.67 | 68 | 17.23 | 68 | 10.37 |
| *GPT-3.5* | 33 | 34.84 | 33 | 18.70 | 33 | 10.96 |
| T5 | **15** | 37.80 | **15** | 19.92 | **15** | 11.5 |
| Ours (init. w/ T5) | 15 | **33.64** | 15 | **18.09** | 15 | **10.66** |

randomly selects a sentence from the retrieved documents. For abstractive compression, we report the performance of the off-the-shelf **T5 (large, 770M)** model and that of **GPT-3.5** model. As we experimented with multiple prompts for the language modeling task, we report the performance of the summaries generated by **GPT-3.5** model with the best single prompt.

**Oracle** We explore the performance upperbound of compression by considering two oracle approaches. For the extractive approach, we construct oracle compressor by considering all sentences $s_i$ in the evidence document set and choosing the sentence that leads to the best end task performance for each example. For the abstractive approach, we consider summaries generated from different prompts ($\{s_j\}_1^n$ in Figure 3) and an empty summary, and choose the one that leads to the best end task performance. As oracle compression is model dependent, we also report model-independent results by using GPT-2 as a reference LM (*Oracle w/ gpt2*) to test how well oracle sentences for one model transfer to other models for the language modeling task.

## 5 RESULTS

**Language modeling** Table 1 reports the results on language modeling task. *All* retrieval augmentation methods improve perplexity over no retrieval setting across three LMs. Heuristic token / phrase-level compression methods (BoW and NE) are worse than prepending uncompressed documents, potentially due to the disfluency of the prepended text.

Both oracle settings show substantial gain over prepending the entire document set, with only 6-13% of tokens. More tokens are not always better: prepending top 1 document outperforms prepending top 5 documents. This confirms that the naive retrieve-and-prepend approach has a significant room for improvement, as prepending irrelevant documents can hurt performances.

Our trained extractive compressor significantly outperforms other extractive baselines (Contriever and BM25) across all three LMs, while prepending slightly fewer tokens. Compared to prepending one document, we achieve a compression ratio of 25% at minimum performance drop. Our trained abstractive compressor performs the best across the board, achieving the lowest perplexity **and** the highest compression ratio. Our abstractive compressor achieves high compression rate through

Table 2: Open-domain QA results with Flan-UL2 (20B) as the LM $M$. We report number of tokens provided as in-context evidence document, excluding the in-context examples. We train separate compressors (one extractive, one abstractive) for each dataset. Extractive compressor selects one sentence for NQ/TQA, and two sentences for HotpotQA.

| In-Context evidence | # tok | NQ EM | F1 | # tok | TQA EM | F1 | # tok | HotpotQA EM | F1 |
|---|---|---|---|---|---|---|---|---|---|
| - | 0 | 21.99 | 29.38 | 0 | 49.33 | 54.85 | 0 | 17.80 | 26.10 |
| *RALM without compression* | | | | | | | | | |
| Top 1 documents | 132 | 33.07 | 41.45 | 136 | 57.84 | 64.94 | 138 | 28.80 | 40.58 |
| Top 5 documents | 660 | **39.39** | **48.28** | 677 | **62.37** | **70.09** | 684 | **32.80** | **43.90** |
| *Phrase/token level compression* | | | | | | | | | |
| Top 5 documents (NE) | 338 | 23.60 | 31.02 | 128 | 54.96 | 61.19 | 157 | 22.20 | 31.89 |
| Top 5 documents (BoW) | 450 | 28.48 | 36.84 | 259 | 58.16 | 65.15 | 255 | 25.60 | 36.00 |
| *Extractive compression of top 5 documents* | | | | | | | | | |
| *Oracle* | 34 | 60.22 | 64.25 | 32 | 79.29 | 82.06 | 70 | 41.80 | 51.07 |
| Random | 32 | 23.27 | 31.09 | 31 | 50.18 | 56.24 | 61 | 21.00 | 29.86 |
| BM25 | 36 | 25.82 | 33.63 | 37 | 54.67 | 61.19 | 74 | 26.80 | 38.02 |
| DPR | 39 | 34.32 | 43.38 | 41 | 56.58 | 62.96 | 78 | 27.40 | 38.15 |
| Contriever | 36 | 30.06 | 31.92 | 40 | 53.67 | 60.01 | 78 | 28.60 | 39.48 |
| Ours | 37 | 36.57 | 44.22 | 38 | **58.99** | 65.26 | 75 | **30.40** | **40.14** |
| *Abstractive compression of top 5 documents* | | | | | | | | | |
| *Oracle* | 51 | 45.68 | 53.66 | 37 | 71.01 | 76.38 | 102 | 35.80 | 46.25 |
| *GPT-3.5* | 56 | 37.12 | 46.35 | 41 | 62.03 | 69.66 | 107 | 31.60 | 42.65 |
| T5 | **10** | 25.90 | 34.63 | **7** | 55.18 | 62.34 | **7** | 23.20 | 33.19 |
| Ours | 36 | **37.04** | **45.47** | 32 | 58.68 | **66.34** | 64 | 28.20 | 37.91 |

selective augmentation, prepending summaries to only 33% of examples (length distribution of generated summaries in Fig. 8).

**Open-domain QA** We report the results on QA tasks in Table 2. Similar to the language modeling task, all retrieval augmentation methods improve performance over no retrieval setting, across three datasets, consistent with previous study on other LMs (Shi et al., 2023b; Mallen et al., 2022; Si et al., 2022). Unlike language modeling, prepending five documents shows significant gains over prepending a single document, motivating the use of compression to incorporate more documents.

We find that extractive oracle outperforms the abstractive one in all datasets. Extractive oracle selects the best one from $N$ candidate sentences, while abstractive oracle selects from two options – prepending GPT-3.5 summary or prepending nothing. Both oracles show improvements over prepending all information, suggesting that removing irrelevant information benefit the model.[8]

Among extractive baselines, DPR performs the best as it has been fine-tuned on high-quality NQ data. On NQ, selecting the top 1 DPR ranked sentences from top 5 documents outperforms prepending top 1 document, with much fewer tokens (39 vs. 132). However, its performance degrades in out of domain datasets. Off-the-shelf summarization model (T5) boasts the highest level of compression, achieving 4-6 points gains in EM while adding mere 7-10 tokens.

The trained compressors, both extractive and abstractive, shows promising performances. On NQ and TQA, the abstractive approach is more effective. On NQ, it achieves a compression ratio of 5% tokens while losing 2 EM points compared to prepending full documents. On TQA, we observe similar trends, compression ratio of 5% tokens while losing 3.7 EM points compared to prepending full sets of documents. On HotpotQA which requires multihop understanding of documents, we find extractive approach to be more helpful, achieving 11% compression rate while losing 2.4 EM points compared to prepending full documents. We find that learning an abstractive compressor for more complex tasks, such as HotpotQA, demands further study. While extreme-scale LLM boasts competitive summarization performance for single document, they are not good at synthesizing information from multiple documents (Shaib et al. (2023); See Section 6 for further analysis).

---

[8]We provide an example where our compressed summary yields correct answer while prepending full document does not in Table 15 in the appendix.

# 6    ANALYSIS AND DISCUSSIONS

**Transferring Across Different LMs**    One benefit of textual summary is that they can transfer to other LMs, unlike approaches such as soft prompts  (Wingate et al., 2022; Chevalier et al., 2023; Mu et al., 2023). We evaluate whether our compressors trained to achieve high performance with respect to a specific LM (GPT2 for language modeling, FlanUL2 for open domain QA) can transfer to other LMs. For language modeling, we find that trained compressor transfers well to other LMs (GPT2-XL and GPT-J), despite they are much larger LMs (Table 1. For open domain QA, we tested transferring our compressors to LLaMA-13B (Touvron et al., 2023) (Table 11 in the appendix). Overall, the performance is worse than the LM from which compressors are trained on, sometimes unable to outperform other compression baselines (e.g., no clear gain from using contriever vs. our trained contriever on TQA/HotpotQA), leaving considerable gap to the oracle compression for LLaMA itself. Yet, on NQ/TQA, our compressor obtains 5% compression ratio with less than 5 EM drop compared to full document setting, showing the robustness of our retrieve-compress-prepend paradigm.

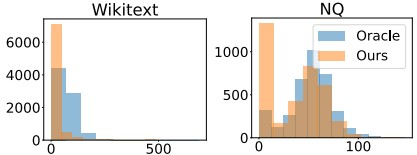

Figure 4: Histogram of abstractive summary length (# tokens) distribution.

**How do the length of the summaries vary?**    Can the learned compressor reliably determine when LMs require retrieved documents? As retrieved documents were **hurting** the model performances for some input queries, 4-24% of training examples for abstractive compressors contain empty summary. Fig. 4 presents the length distribution of abstractive summaries on NQ and Wikitext (histograms for other datasets is in Fig. 8 in the appendix). The input document lengths do not vary significantly across examples, yet we find the abstractive summary vary significantly in length, suggesting abstractive compressor enables selective retrieval augmentation. We have not experimented selective compression with extractive compressor, fixing the number of prepended sentences for the entire dataset. We leave enabling adaptive augmentation with extractive summarizer to future work.

**How does model leverage the in-context documents?**    We evaluate whether retrieval augmented LMs tend to copy answers verbatim from in-context evidence documents or generate answers not present in the documents. This is an desired behavior *only* when the gold answer is in the evidence. We first report how frequently a gold answer span is present in evidence text (% Gold in Evi). As expected, full documents contain the answer most frequently, followed by NE and GPT-3 compression. However, having more gold answers in the evidence doesn't equate to better performance, as the model cannot always identify the correct answer from the evidence (84 % for **NE** v.s. 98% for **T5(ours)**).

We also observe that model can be easily distracted by irrelevant contexts, copying from a document span even when it does not contain gold answer, echoing findings from prior work (Shi et al., 2023a). Prepending top 5 documents has a higher frequency (81%) of copying incorrectly compared to top 1 document (51%), and GPT-3 compression leads to an even higher incorrect copying frequency (85%), potentially as query-focused summarization generates sentences that seemingly contains the answer. Our compressor successfully reduce such erroneous behavior to 39%.

Table 3: Analysis on usage of in-context evidence for NQ. For the last column, we report frequency of model copying from the evidence on a subset where gold answer is present / when it is not.

| Evidence | EM | %Gold in Evi. | %Pred in Evi. |
|---|---|---|---|
| Top 1 | 33.1 | 36 | 92 / 51 |
| Top 5 | 39.3 | **57** | 96 / 81 |
| NE | 26.0 | 46 | 84 / 48 |
| Oracle sent | 60.2 | 34 | 93 / 16 |
| Contriever | 30.2 | 25 | 88 / 36 |
| Ours | 36.6 | 28 | 90 / 33 |
| GPT-3.5 | 37.1 | 45 | **98 / 85** |
| T5 | 25.9 | 30 | 52 / 20 |
| Ours | 37.0 | 34 | **98** / 39 |

**Are the generated summaries faithful and comprehensive?**    We (the authors) manually evaluate outputs of the abstractive compressors on two axes  (Chen et al., 2023): **Faithfulness**: whether the summary can be entailed by the retrieved documents, **Comprehensiveness**: whether the summary contains sufficient information to answer the question, regardless of whether the generated information comes from the retrieved documents. For both, we select one of three labels: **Y**es, **P**artially, **No**, and report the % of **Useful** summaries which are both faithful and comprehensive. Annotation sample

can be found in Table 16 in the appendix. We evaluate the summaries generated by GPT-3.5 and our abstractive compressor. We randomly sample 30 non-empty summaries from the test set.

Table 4 presents annotation results. GPT-3.5, substantially bigger than our compressor, generates more useful summary across all three datasets. Overall, our abstractive compressors were less faithful compared to the original GPT-3.5, while improving comprehensiveness. The effectiveness of summarization also depends on the datasets – summaries from both models were the most faithful for TQA and the least faithful for HotpotQA dataset. In terms of comprehensive-ness, we find both models easily find the information for NQ, but struggle with HotpotQA. These results partially explain why the performance gain was limited for HotpotQA.

Table 4: Manual analysis on abstractive summaries generated for NQ, TQA and HotpotQA (HQA) dataset.

| Dataset | Model | % Faithful | | | % Compre. | | | % Use. |
|---|---|---|---|---|---|---|---|---|
| | | Y | P | N | Y | P | N | |
| NQ | GPT-3.5 | 90 | 0 | 10 | 97 | 0 | 3 | 83 |
| | Ours | 80 | 13 | 7 | 100 | 0 | 0 | 80 |
| TQA | GPT-3.5 | 97 | 0 | 3 | 90 | 0 | 10 | 83 |
| | Ours | 83 | 3 | 14 | 96 | 0 | 4 | 77 |
| HQA | GPT-3.5 | 74 | 0 | 26 | 78 | 0 | 22 | 50 |
| | Ours | 67 | 0 | 33 | 74 | 0 | 26 | 40 |

## 7    RELATED WORK

**Efficient RALM**    He et al. (2021) improves efficiency of RALMs by reducing retrieval cost, such as data store compression and dimensionality reduction for neural retriever. A line of work proposes to reduce retrieval frequency (He et al., 2021; Mallen et al., 2022; Martins et al., 2022). In this work, we improve efficiency of RALM by compressing retrieved documents into a concise summary or an empty sequence, facilitating selective retrieval augmentation.

**Prompt Compression**    Recent work (Wingate et al., 2022; Chevalier et al., 2023; Mu et al., 2023) proposes compressing long contexts into soft prompts that can be used by LMs, rather than textual summaries. Such soft prompts can serve as efficient replacements for plain-text context, minimizing the computational costs during inference. Another related line of work proposes context distillation (Snell et al., 2022; Choi et al., 2022; Padmanabhan et al., 2023), which injects the prepended context into the parameters of an LM. Compared to above approaches, our approach yields more interpretable textual summary that can transfer across different LMs, and can be applied to black box LMs without requiring gradient updates. Prior work has studied textual compression for other tasks, such as political fact checking (Chen et al., 2023) and instruction learning (Yin et al., 2023). Concurrent works (Li et al., 2023; Jiang et al., 2023) propose token-level prompt compression methods which leverage a small model to prune out redundant tokens, while our method preserves readability of the compressed text. Wang et al. (2023) proposes to filter irrelevant retrieved documents and Yoran et al. (2024) finetunes the language model to be able to properly handle irrelevant contexts.

**Distillation / Goal Oriented Summarization**    Recent work introduces symbolic knowledge distilla-tion (West et al., 2022), which transfers knowledge from a teacher model by training a student model on a training dataset generated with the teacher model. For better performance, they introduce critic criteria, which filter undesirable examples from generated training dataset. Such distillation technique has been applied for various applications including summarization (Jung et al., 2023), which aims to generate high quality summaries while we optimize for generating effective summary for downstream LMs. Our setting is similar to Hsu & Tan (2021), which trains an extractive summarization model to optimize for prediction accuracy of a sentiment prediction model which uses the summary.

## 8    CONCLUSION

We introduce **RECOMP**, a method which compresses retrieved documents into textual summaries before prepending them to improve in-context retrieval augmented language models. We present two compression models – an extractive compressor and an abstractive compressor. We design a training scheme which leverages end task signals from a blackbox LM to generate useful summaries and enable selective augmentation. Our experiments show that our compressors can improve the efficiency of retrieval augmented LMs significantly with minimal drop in performances.

## ACKNOWLEDGEMENT

We thank the members of the UT and UW NLP community for feedback on the project. We especially thank Alisa Liu, Junyi Jessy Li and Greg Durrett for providing comments on the draft. The project is partially funded by NSF grant (IIS-2312948).

## ETHICS STATEMENT

We use commercial language model to generate training data for our compressors, which might include factual error. We conduct careful human evaluation on the data generated and present our analysis in the paper.

## REPRODUCIBILITY STATEMENT

We release our codes, prompt, and data generated with API access publicly.

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

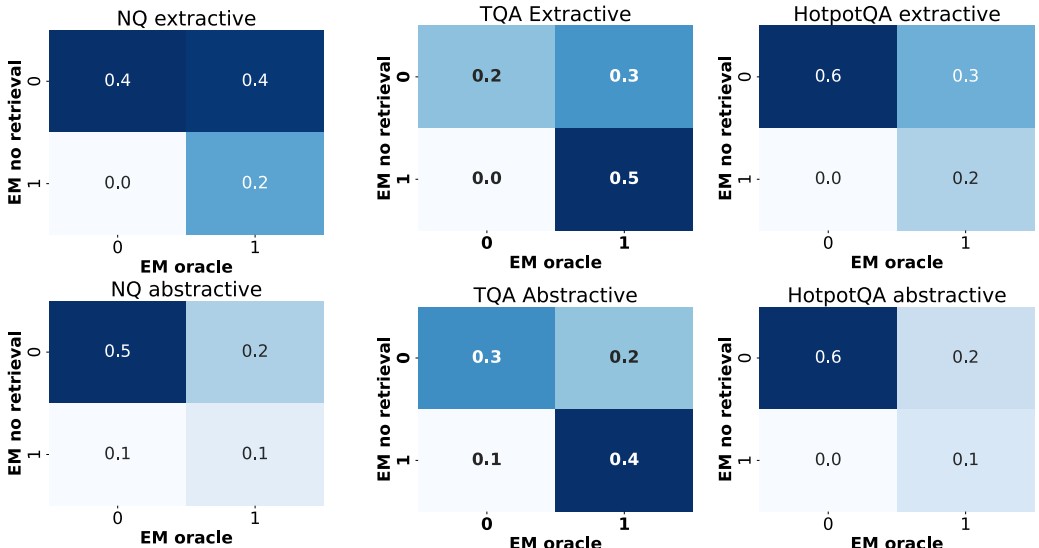

Figure 5: We report the data distribution on NQ dev set, TriviaQA dev set and HotpotQA dev set comparing the end task performance when prepending the oracle compression method (oracle sentence or GPT-3 summaries) and when not prepending anything for the base model (Flan-UL2).

## A  APPENDIX

### A.1  COMPRESSOR TRAINING DATA GENERATION

We report the statistics of the data used to train compressors in Table 5. We use SpaCy (Honnibal et al., 2020) to extract named entities.

**Extractive Data Generation**  We generate data using the training data for the four datasets we tested (Wikitext, NQ, TQA and HotpotQA). We use the NLTK package to perform sentence splitting. We remove examples without any negatives.

**Abstractive Data Generation**  We report prompt used to generate summaries in Table 14. We queried the Open AI API with temperature of 0.7 and top p = 1. For the language modeling task, we use an ensemble of four prompts and choose the one which leads to the lowest perplexity as the target. If none of the summaries lead to perplexity decrease, we treat an empty summary as target. We queried the OpenAI API with temperature of 0.7 and top p = 1. We generate four summaries per example for randomly sampled 2% of the training data (48,013 examples).

### A.2  COMPRESSOR TRAINING DETAILS

**Extractive Compressor**  For language modeling, we use the contriever checkpoint [9] trained with unsupervised data. For the QA tasks, we use the contriever checkpoint fine-tuned on the MSMARCO task (Campos et al., 2016) [10] , following prior work (Si et al., 2022; Shi et al., 2023b). We implement the model using the Transformers (Wolf et al., 2019) and the sentence-transformer library (Reimers & Gurevych, 2019). We train with Adam optimizer (Kingma & Ba, 2014), using a batch size of 64, learning rate of 2e-5 and 1000 warmup steps for 3 epochs. We report results on the model with the best reranked perplexity on our validation set for the language modeling task and the best reranked accuracy for the QA tasks.

---

[9] https://huggingface.co/facebook/contriever
[10] https://huggingface.co/facebook/contriever-msmarco

Table 5: Training data statistics for abstractive and extractive compressors. "% filtered" refers to the percentage of examples not included. For extractive compressor, we filter examples for which prepending sentences with top contriever score leads to the correct answer. For abstractive compressor, we filter examples for which prepending GPT-3.5-turbo generated summary and no retrieval both produce incorrect answers. $|\mathcal{N}|$ refers to the average number of negative examples for constrastive training.

| Dataset | Extractive | | | | Abstractive | | | |
| | Train | Validation | % filtered | $|\mathcal{N}|$ | Train | Validation | % filtered | % empty |
| --- | --- | --- | --- | --- | --- | --- | --- | --- |
| NQ | 42,149 | 9,769 | 46 | 4.44 | 39,466 | 4,931 | 50 | 25 |
| TQA | 70,032 | 8,753 | 56 | 4.37 | 48,322 | 5,887 | 32 | 16 |
| HotpotQA | 24,526 | 3,068 | 69 | 4.33 | 26,556 | 2,937 | 42 | 4 |
| Wikitext | 1,398,318 | 1,5483 | 41 | 4.04 | 38,410 | 9,603 | 0 | 24 |

Table 6: Inference speed of FLAN-UL2 on NQ dev set.

| Setting | # toks | Inference time | Compression time | Total time | # example / s |
| --- | --- | --- | --- | --- | --- |
| No docs | 0 | 4,104s | 0s | 4,104s | 0.88 |
| Top 5 docs | 660 | 10,584s | 0s | 1,0584s | 0.34 |
| Contriever (ours) | 37 | 4,880s | 113s | 4,993s | 0.72 |
| T5 (ours) | 36 | 4,936s | 871s | 5,807s | 0.62 |

**Abstractive Compressor** We implement the model using the Transformers (Wolf et al., 2019). We train abstractive summarizer with Adam optimizer (Kingma & Ba, 2014), using a batch size of 16, learning rate of 1e-5 and 1000 warmup steps for 3 epochs.

## A.3 INFERENCE SPEED

We report the number of tokens as a measure for efficiency in the main paper. Here, we report inference speed measured by GPU time as an additional metric for efficiency. Specifically, we report the GPU time for NQ dev set with FLAN-UL2 as the base model. We run FLAN-UL2 on 4 A40 GPUs. For compression, we run contriver and T5 on a single A40 GPU (Table 6). Our methods, even when including the time taken for compressor inference, improve throughput compared to prepending the full sets of documents, with contriever being more efficient, enabling 2x throughput.

We note that unlike the number of tokens, inference speed is contigent on implementation and size of the base model. For instance, bigger models will suffer from more latency with more input tokens and thus RECOMP can bring more speed-up.

Table 7: Compressing retrieved documents for contriever and DPR on NQ.

| Doc | Compression | DPR | | Contriever | |
| | | # tokens | EM | # tokens | EM |
| --- | --- | --- | --- | --- | --- |
| No | - | 0 | 21.99 | 0 | 21.99 |
| Top 1 | - | 133 | 36.59 | 130 | 27.53 |
| Top 5 | - | 667 | *42.44* | 652 | *33.19* |
| Top 5 | Contriever | 26 | 25.60 | 31 | 23.55 |
| Top 5 | Extractive (ours) | 36 | 35.76 | 35 | **30.66** |
| Top 5 | T5 | 10 | 26.84 | 10 | 24.32 |
| Top 5 | Abstractive (ours) | 35 | **37.40** | 28 | 30.55 |

Table 8: Comparing prepending top 5 documents and prepending GPT-3.5-turbo compressed documents.

| GPT 3.5 EM | Top 5 EM | % Data | | |
|---|---|---|---|---|
| | (no compression) | NQ | TQA | HotpotQA |
| 0 | 0 | 53 | 31 | 61 |
| 0 | 1 | 10 | 7 | 8 |
| 1 | 0 | 7 | 7 | 6 |
| 1 | 1 | 30 | 55 | 25 |

### A.4 COMPRESSING DOCUMENTS RETRIEVED BY DIFFERENT RETRIEVAL MODELS

We experiment with using our compressor trained with one retrieval model to compress texts retrieved by other retrievers. We report results on NQ (Table 7, compressing documents retrieved by contriever and DPR with compressors trained with contriever-ms-marco. We see that both our extractive and abstractive compressor generalize robustly to documents retrieved by another retrieval system, with the best compressor outperforming prepending top 1 documents with 4x compression.

### A.5 MORE ANALYSIS ON ABSTRACTIVE COMPRESSOR FOR THE QA TASKS

#### A.5.1 IS THE TEACHER MODEL A GOOD SUMMARIZER?

As shown in Table 2, compressing with the teacher model (GPT-3.5-turbo) results in slight performance drop compared to prepending all top 5 documents for all three datasets (NQ, HotpotQA and TQA). We look closer into where the performance gap comes from by comparing the end task performance (Exact Match) between prepending GPT-3.5-turbo generated summaries and prepending the top 5 documents (Table 8). Compressing with GPT-3.5-turbo leads to performance decreases for 10%, 7% and 8% of data on NQ, TQA and HotpotQA. Notably, it also leads to an increase for a small portion of data (7%, 7% and 6%) on the three datasets.

**Manual analysis** We randomly sample 20 summaries for which compressing with GPT-3.5-turbo leads to different performance than prepending top 5 documents per datasets to conduct manual analysis. We report manual analysis results in Table 9. **Ambiguous questions** refer to the cases where the question itself is ambiguous (Min et al., 2020) and answers from both systems are valid. **Semantic equivalence** are cases where both answers are semantically similar. As we can see, these two

Table 9: Manual analysis of when compressing with GPT-3.5-turbo leads to different performance compared to prepending top 5 documents. We report % examples belonging to each reason.

| Reason | NQ | TQA | HotpotQA |
|---|---|---|---|
| Ambiguous question | 17 | 11 | 5 |
| Semantic equivalence | 44 | 32 | 15 |
| Insufficient evidence | 11 | 42 | 50 |
| Lost in the middle | 11 | 10 | 15 |
| Incorrect summary | 17 | 5 | 15 |

categories constitute a non-trivial portion of the data analyzed. **Insufficient evidence** refers to the cases where the retrieved documents cannot sufficiently support the gold answer. This can include cases where (1) GPT-3.5-turbo hallucinates the gold answer or (2) when prepending top 5 documents, the model generates the correct answer, yet it cannot be fully supported by the retrieved documents. This also demonstrates headroom in improving the retrieval system, especially for more challenging dataset such as HotpotQA. **Lost in the middle** refers to cases where the model is unable to extract the correct answer from the top 5 documents while compression helps. Finally, **Incorrect summary** refers to genuine errors caused by compressing the retrieved documents.

#### A.5.2 DISTILLATION PERFORMANCE ANALYSIS

We examine whether the abstractive compressors successfully distill the teacher model on the three QA datasets. We compare performances of FLAN-UL2 when compressing with our T5 model and when compressing with the oracle GPT-3.5-turbo model (Table 10. We see that models face

Table 10: Analysis on distillation performance, grouped by whether the generated summary is an empty summary. The left two column denotes whether the summaries generated by T5 or oracle GPT-3.5-turbo is empty (**E**) or not (**N**). We report the **EM** for the T5 system and the oracle system (**EM(O)**) and the percentage of data belong to this slice.

| T5 | O | NQ | | | TQA | | | HotpotQA | | | Comment |
|---|---|---|---|---|---|---|---|---|---|---|---|
| | | EM | EM(O) | % | EM | EM(O) | % | EM | EM(O) | % | |
| N | N | 0.47 | 0.49 | 60 | 0.68 | 0.76 | 70 | 0.30 | 0.34 | 85 | Both generate a summary. |
| N | E | 0.31 | 1.0 | 4 | 0.26 | 1.0 | 6 | 0.12 | 1.0 | 3 | Incorrect selective augmentation. |
| E | N | 0.11 | 0.25 | 32 | 0.32 | 0.41 | 21 | 0.17 | 0.24 | 11 | |
| E | E | 1.0 | 1.0 | 4 | 1.0 | 1.0 | 3 | 1.0 | 1.0 | 1.0 | T5 correctly generates an empty summary. |

challenges in performing selective distillation for all three datasets. Our systems fail at performing selective augmentation for 36%, 27% and 14% of the data on NQ, TQA and HotpotQA (it either attempts to generate a summary when the teacher model fails to do so, or outputs an empty summary when generating a summary will lead to better performance). When both systems generate a summary, our distillation is relatively successful on NQ (4% performance drop) but less effective on TQA (11%) and HotpotQA (12%) performance drop.

**Selective augmentation** We further calculate performance of selective augmentation against the oracle method, i.e. we treat selective augmentation as a binary classification task and evaluate whether our compressor correctly chooses to output an empty summary (or not). The F1 performance is 0.19, 0.20 and 0.10 for NQ, TQA and HotpotQA respectively. The performance of a random baseline, which randomly outputs an empty summary at the same frequency of the oracle method, is 0.10, 0.08, and 0.00 for NQ, TQA and HotpotQA. This shows that although our compressors learn to perform selective augmentation to some extent, there is significant headroom to improve selective augmentation performance.

### A.5.3 TRANSFERRING LEARNED COMPRESSORS TO ANOTHER MODEL FOR QA TASKS

We report results of transferring our compressors to LLaMA-13B (Touvron et al., 2023) model in Table 11. We use exactly the same input (e.g. same in-context example and prepended documents) as the FLAN-UL2 model. We report oracle for the FLAN-UL2 model (*Oracle (w/ FLAN)*) as well as oracle for LLaMA-2. While we see successful transfer for the language modelling task from GPT2 to GPT2-XL and GPT-J, transferring compressors trained for QA tasks to another model yields rather negative results. First, we see that oracle compression for FLAN-UL2 model significantly lags behind the oracle method for LLaMA-2 for both compression methods and all three datasets. As oracle method depends on parametric knowledge of the model (i.e. the model relies on parametric when an empty summary is prepended), one potential reason for transfer failure is that compression model trained with one model might not transfer to another model with parametric knowledge differences. We also note that for summaries generated by GPT-3.5 on HotpotQA, which work almost as well as prepending top 5 documents for FLAN-UL2, lags behind prepending top 5 documents when prepended to LLaMA. This shows that summaries which can be effectively consumed by one model might not be able to transfer to another model. Future work should look into how to build compression model that work across different models, especially for downstream tasks such as question answering.

Table 11: Open-domain QA results on LLaMA-13B. We report the results of oracle compressions with Flan-UL2, the base model for the compressors, (*Oracle w/ FLAN*) and the oracle compression results for LLaMA-13B.

| In-context evidence | # tok | NQ EM | F1 | # tok | TQA EM | F1 | # tok | HotpotQA EM | F1 |
|---|---|---|---|---|---|---|---|---|---|
| - | 0 | 30.89 | 40.73 | 0 | 65.00 | 71.18 | 0 | 24.20 | 34.50 |
| *RALM without compression* | | | | | | | | | |
| Top 1 document | 132 | 33.35 | 43.13 | 136 | 66.62 | 73.10 | 138 | 34.40 | 44.17 |
| Top 5 documents | 660 | 37.04 | 47.60 | 667 | 70.61 | 77.51 | 684 | 37.00 | 47.11 |
| *Phrase / token level compression* | | | | | | | | | |
| Top 5 documents (BoW) | 450 | 33.05 | 43.36 | 259 | 66.59 | 73.40 | 255 | 30.00 | 39.13 |
| Top 5 documents (NE) | 338 | 34.60 | 44.91 | 128 | 65.88 | 72.59 | 157 | 29.20 | 37.93 |
| *Extractive compression of top 5 documents* | | | | | | | | | |
| *Oracle* | 31 | 56.62 | 68.89 | 31 | 84.61 | 80.46 | 69 | 42.20 | 51.34 |
| *Oracle (w/ FLAN)* | 34 | 40.89 | 50.06 | 32 | 68.52 | 74.96 | 70 | 35.20 | 45.13 |
| Random | 32 | 30.33 | 39.85 | 31 | 62.80 | 69.25 | 61 | 27.40 | 36.27 |
| Contriever | 36 | 32.52 | 42.01 | 40 | 65.88 | 72.44 | 78 | 34.60 | 43.99 |
| Ours (init. w/ Contriever) | 37 | 34.38 | 44.15 | 38 | 65.28 | 71.85 | 75 | 33.20 | 42.88 |
| *Abstractive compression of top 5 documents* | | | | | | | | | |
| *Oracle* | 50 | 45.60 | 84.87 | 38 | 74.37 | 79.83 | 98 | 41.40 | 51.54 |
| *Oracle (w/ FLAN)* | 51 | 38.98 | 49.40 | 37 | 69.86 | 76.46 | 102 | 35.40 | 46.17 |
| *GPT-3.5* | 56 | 35.71 | 46.05 | 41 | 67.90 | 74.88 | 107 | 34.60 | 45.40 |
| T5 | 10 | 33.38 | 43.54 | 7 | 63.18 | 70.92 | 7 | 30.40 | 40.60 |
| Ours (init. w/ T5) | 36 | 36.32 | 46.10 | 32 | 66.27 | 73.12 | 81 | 30.80 | 40.61 |

---

**Input:** Base LM $M$, Compressor encoder $\mathtt{enc}_\theta$, Training data $\{\mathbf{x_i}, \mathbf{S_i}, \mathbf{y_i}\}_1^T$ where $\mathbf{x_i}$ is input, $\mathbf{S_i} = \{\mathbf{s_j}\}_1^n$ is a set of candidate sentences from the retrieved document for $\mathbf{x_i}$, $\mathbf{y_i}$ is the target answer.
**Output:** An updated extractive compressor encoder $\mathtt{enc}_\theta$
1: $\mathcal{T} \leftarrow \emptyset$
2: **for** $i \in \{1, \ldots, T\}$ **do**
3:     $\mathbf{p_i} \leftarrow \mathrm{argMax}_{\mathbf{s_j} \in \{\mathbf{S_i}\}}\mathbf{Score}(M, \mathbf{y_i}, [\mathbf{s_j}; \mathbf{x_i}])$
4:     **for** $j \in \{1, \ldots, n\}$ **do**
5:        $\mathcal{L} \leftarrow \emptyset$
6:        **if** $\mathbf{Score}(M, \mathbf{y_i}, [\mathbf{s_j}; \mathbf{x_i}]) < \mathbf{Score}(M, \mathbf{y_i}, [\mathbf{p_i}; \mathbf{x_i}])$ **then**
7:           $\mathcal{L} \leftarrow \mathcal{L} \cup \mathbf{s_j}$
8:     **if** $|\mathcal{L}| > 0$ **then**
9:        $\mathcal{N}_i \leftarrow \mathrm{argTop5}_{\mathbf{s_j} \in \mathcal{L}}(\langle \mathtt{enc}_\theta(\mathbf{s_j}), \mathtt{enc}_\theta(\mathbf{x_i}) \rangle)$
10:     $\mathcal{T} \leftarrow \mathcal{T} \cup \{(\mathbf{x_i}, \mathbf{p_i}, \mathcal{N}_i)\}$
11: $\mathtt{enc}_\theta = \mathbf{Finetune}(\mathtt{enc}_\theta, \mathcal{T})$

Figure 6: Learning an extractive compressor for QA task. The **Score** here is the exact match between the decoded answer and the gold answers.

**Input:** Teacher LM $M_t$, Base LM $M$, Summarization prompt $p$, Compressor $\texttt{encdec}_\theta$, Training data $\{\mathbf{x_i}, \mathbf{D_i}, \mathbf{y_i}\}_1^T$ where $\mathbf{x_i}$ is input, $\mathbf{D_i}$ is the set of retrieved document for $\mathbf{x_i}$, $\mathbf{y_i}$ is the target answer.
**Output:** An updated $\texttt{encdec}_\theta$

```
 1: T ← ∅
 2: for i ∈ {1, . . . , T} do
 3:     s_i = Decode(M_t, [p; x_i; D_i])
 4:     v_s = Score(M, y_i, [s_i; x_i])
 5:     v_d = Score(M, y_i, [x_i])
 6:     if v_s < v_d then
 7:         T ← T ∪ {(x_i, D_i, ∅)}
 8:         continue
 9:     T ← T ∪ {(x_i, D_i, s_i)}
10: encdec_θ = Finetune(encdec_θ, T)
```

Figure 7: Learning an abstractive compressor for QA task. The **Score** here is the exact match between the decoded answer and the gold answers.

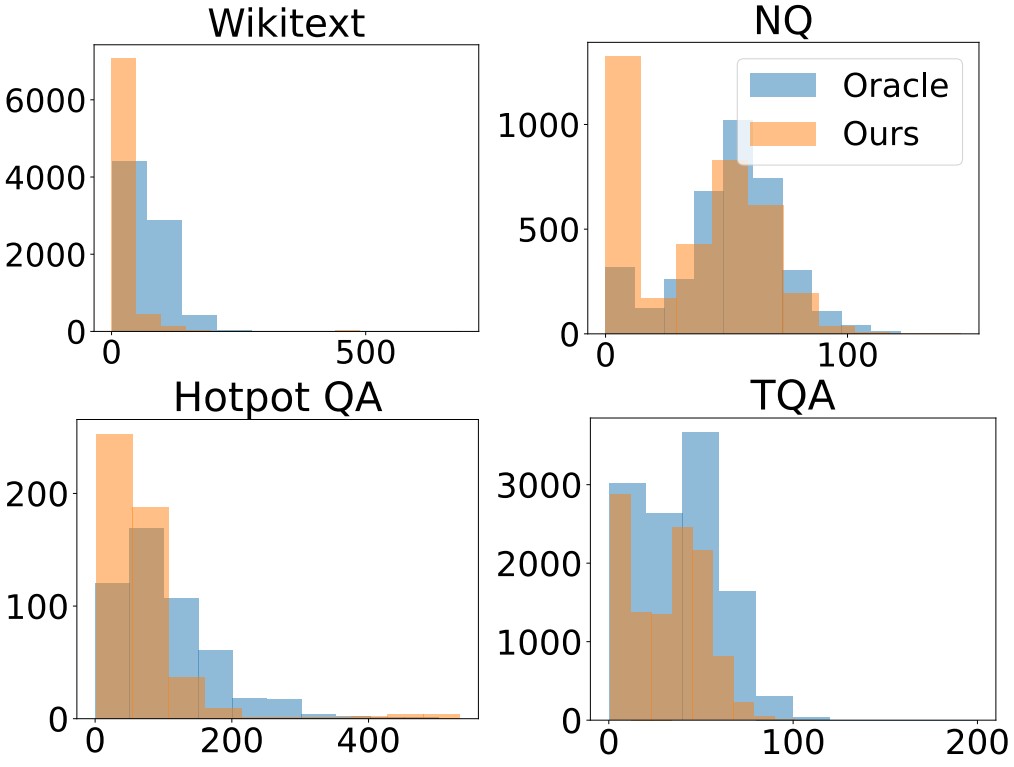

Figure 8: Histogram of abstractive summary length (# tokens) distribution for testing data of NQ,TQA, HotpotQA and Wikitext.

Table 12: Example abstractive and extractive compression on wikitext-103 dev set and NQ.

| **Wikitext-103 Input** present in most of the Mediterranean Sea, only missing from the section east of Crete, and along only the north @-@ west coast of the Black Sea | **Original Top 1 document** Sea of Crete" Sea of Crete The Sea of Crete (, ""Kritiko Pelagos"") or Cretan Sea, is a sea, part of the Aegean Sea, located in its Southern extremity. The sea stretches to the North of the island of Crete, East of the islands of Kythera and Antikythera, South of the Cyclades, and West of the Dodecanese islands of Rhodes, Karpathos and Kassos. The bounding sea to the West is the Ionian Sea. To the Northwest is the Myrtoan Sea, a subdivision of the Mediterranean Sea that lies between the Cyclades and Peloponnese. To the East-SE is the rest of the Mediterranean Sea, |
|---|---|
| **Method** | **Compressed document** |
| BoW | present Mediterranean Sea missing section east Crete along north west coast Black The Kritiko Pelagos Cretan sea part Aegean located Southern extremity stretches North island East islands Kythera Antikythera South Cyclades West Dodecanese Rhodes Karpathos Kassos bounding Ionian To Northwest Myrtoan subdivision lies Peloponnese SE rest |
| NE | Kythera the Aegean Sea the Ionian Sea Crete Southern South Rhodes the Myrtoan Sea Cretan Sea Antikythera Dodecanese Kassos Karpathos West the Black Sea & Sea of Crete Peloponnese the Mediterranean Sea Cyclades |
| Extractive compression | To the Northwest is the Myrtoan Sea, a subdivision of the Mediterranean Sea that lies between the Cyclades and Peloponnese. |
| **NQ Input** who got the first nobel prize in physics | **Original Top 5 document** receive a diploma, a medal and a document confirming the prize amount. Nobel Prize in Physics The Nobel Prize in Physics () is a yearly award given by the Royal Swedish Academy of Sciences for those who have made the most outstanding contributions for mankind in the field of physics. It is one of the five Nobel Prizes established by the will of Alfred Nobel in 1895 and awarded since 1901; the others being the Nobel Prize in Chemistry, Nobel Prize in Literature, Nobel Peace Prize, and Nobel Prize in Physiology or Medicine. The first Nobel Prize in Physics was science, Ernest Lawrence won the Nobel Prize in Physics in 1939. Lars Onsager won the 1968 Nobel Prize in Chemistry. Norman Borlaug, father of the Green Revolution, won the Nobel Peace Prize in 1970. Christian B. Anfinsen won the Nobel Prize for chemistry in 1972. Ivar Giaever won the Nobel Prize in Physics 1973. Carl Richard Hagen is noted for his work in physics. In engineering, Clayton Jacobson II is credited with the invention of the modern personal watercraft. Ole Singstad was a pioneer of underwater tunnels. Ole Evinrude invented the first outboard motor with practical commercial application, recognizable today Nobel Prize in Physics The Nobel Prize in Physics () is a yearly award given by the Royal Swedish Academy of Sciences for those who have made the most outstanding contributions for mankind in the field of physics. It is one of the five Nobel Prizes established by the will of Alfred Nobel in 1895 and awarded since 1901; the others being the Nobel Prize in Chemistry, Nobel Prize in Literature, Nobel Peace Prize, and Nobel Prize in Physiology or Medicine. The first Nobel Prize in Physics was awarded to physicist Wilhelm R00f6ntgen in recognition of the extraordinary services he was also awarded the Abel prize. In addition, eight ñormaliens have gone on to receive the Nobel Prize in Physics: Claude Cohen-Tannoudji, Pierre-Gilles de Gennes, Albert Fert, Alfred Kastler, Gabriel Lippmann, Louis N 00e9el, Jean Baptiste Perrin and Serge Haroche, while other ENS physicists include such major figures as Paul Langevin, famous for developing Langevin dynamics and the Langevin equation. Alumnus Paul Sabatier won the Nobel Prize in Chemistry. A ranking of universities worldwide based on ratios of alumni to Nobel prize-winners published in 2016 by American scholars Stephen Hsu and Jonathan Wai placed ENS as the first university worldwide, far rendered by the discovery of the remarkable rays (or x-rays). This award is administered by the Nobel Foundation and widely regarded as the most prestigious award that a scientist can receive in physics. It is presented in Stockholm at an annual ceremony on 10 December, the anniversary of Nobel's death. Through 2018, a total of 209 individuals have been awarded the prize. Only three women (1.4% of laureates) have won the Nobel Prize in Physics: Marie Curie in 1903, Maria Goeppert Mayer in 1963, and Donna Strickland in 2018. Alfred Nobel, in his last will and testament, stated that his |
| **Method** | **Compressed document** |
| T5 | Wilhelm Röntgen received the first Nobel Prize in Physics in recognition of his extraordinary services. It is one of the five Nobel Prizes established by Alfred Nobel in 1895 and awarded since 1901. |
| GPT-3.5-turbo | The first Nobel Prize in Physics was awarded to physicist Wilhelm Röntgen in 1901 for his discovery of the remarkable rays (or x-rays). Since then, 209 individuals have been awarded the prize, with only three women (1.4% of laureates) having won it. |

Table 13: Example input to the Flan-UL2 for NQ with in-context examples and retrieved documents.

| Dataset | Prompts |
|---|---|
| NQ | who won a million on deal or no deal Answer: Tomorrow Rodriguez who is the woman washing the car in cool hand luke Answer: Joy Harmon who is the actor that plays ragnar on vikings Answer: Travis Fimmel who said it's better to have loved and lost Answer: Alfred , Lord Tennyson name the first indian woman to be crowned as miss world Answer: Reita Faria Retrieved Docs Question Answer: |

Table 14: Prompts used to generated summaries from GPT-3.5-turbo. `query` and `docs` represent the actual input query and retrieved documents.

| Dataset | Prompts |
|---|---|
| NQ | Compress the information in the retrieved documents into a 2-sentence summary that could be used to answer the question: Question: `query` Retrieved documents: `docs` Compressed documents: |
| TQA | Compress the information in the retrieved documents into a 2-sentence summary that could be used to answer the question: Question: `query` Retrieved documents: `docs` Compressed documents: |
| HotpotQA | Source documents: `docs` Question: `query` Generate a reasoning chain to answer the question: |
| Wikitext | Generate the next two sentences of the given query using the information from the provided documents. \nSource Documents: `docs` \n**Query:** `query` \n |
| Wikitext | Select sentences from the retrieved docs that are most likely be in the next sentence.\nSource Documents: `docs` \n**Query:** `query`\n |
| Wikitext | Generate the next one sentence of the given query using the information from the provided documents\nSource Documents: `docs` \n**Query:** `query` \n |
| Wikitext | Summarize the information from the provided documents\nSource Documents: `docs` \n**Query:** `query`\n |

Table 15: Case study of how compressing the retrieved documents helps the model to identify the right answer from NQ dev set.

| | | |
|---|---|---|
| ***Question:*** *host of the late show who was once a correspondent for the daily show.* **Gold answer**: *Stephen Colbert* | | |
| **Type** | **In-context documents** | **Predicted Answers** |
| None | | Chelsea Handler |
| Top 5 | by Conan O'Brien, in 2009. Leno explained that he did not want to see a repeat of the hard feelings and controversy that occurred when he was given the show over David Letterman following Carson's retirement in 1992. O'Brien's last "Late Night" episode was taped on February 20, 2009. Former Saturday Night Live alum Jimmy Fallon took over as host of "Late Night with Jimmy Fallon" on March 2, 2009. The Colbert Report that aired four days a week on Comedy Central from October 17, 2005, was hosted by Stephen Colbert, one of the regulars on Comedy Central's The Daily season as host began with a notable interview with former British prime minister Tony Blair. The live interview occurred the night before a book signing at Eason's which attracted international attention when Blair was pelted with shoes and eggs and successfully evaded an attempted citizen's arrest on charges of war crimes. On 1 February 2013, Pat Kenny returned to host that night's edition when Tubridy's father died. In 2015, Tubridy's tone and choice of questions when interviewing Anti-Austerity Alliance TD Paul Murphy in relation to the campaign against the implementation of a water tax was much criticised. Opponents of the 'Michigan, interviewing Eminem. Colbert has been given near-full control of the show, with little interference from CBS management in regard to format. Colbert brought most of his staff from "The Colbert Report" with him to "The Late Show", as well as outsiders such as Brian Stack, who is best known for his work on Conan O'Brien's programs, and Jon Stewart, former host of Colbert's previous sister program "The Daily Show", who is credited as executive producer. Colbert no longer uses the character he had portrayed on "The Colbert Report", jokingly remarking to Jeb Bush that "I used to play a Show" has had three regular hosts: Gay Byrne, Pat Kenny and Ryan Tubridy. Frank Hall deputised for Byrne for one season in the 1960s. There have been at least four occasions on which another presenter has hosted the show. The first was when Byrne became unexpectedly and seriously ill. Frequent panelist Ted Bonner presented instead. The second time was towards the end of a show about feminism, when Byrne ushered a young Marian Finucane into his seat to present the remainder of the show. On another occasion, radio broadcaster and former news reader Andy O'Mahony replaced Byrne for an interview popular acclaim. Colbert would host the program until he was chosen to replace David Letterman as host of CBS' "Late Show" in 2015. Ed Helms, a former correspondent from 2002 to 2006, also starred on NBC's "The Office" and was a main character in the 2009 hit "The Hangover". After filling in as host during Stewart's two-month absence in the summer of 2013, John Oliver went on to host his own show on HBO, "Last Week Tonight with John Oliver". In 2016, former correspondent Samantha Bee launched her own late-night talk show "Full Frontal with Samantha Bee". Bee's husband Jason | Samantha Bee |
| GPT-3.5-turbo | Former Daily Show correspondent Stephen Colbert was chosen to replace David Letterman as host of CBS's "Late Show" in 2015, while Ed Helms, a former correspondent from 2002 to 2006, also starred on NBC's "The Office" and John Oliver, who filled in as host during Jon Stewart's absence in 2013, went on to host his own show on HBO. | |
| T5 (ours) | Stephen Colbert was a former correspondent for The Daily Show and later became host of CBS's "Late Show" in 2015. He has since brought most of his staff from "The Colbert Report" with him to "The Late Show", with little interference from CBS management in regard to format. | Stephen Colbert |

Table 16: Example summaries and their manual analysis labels. See Table 17 for more example.

| Dataset | Model | Query, Passages and Summary | Evaluation |
|---|---|---|---|
| NQ | Ours | **Question:** when will miraculous ladybug season 2 episode 12 come out
**Passages:** 2016 on TVNZ's TV2. In Japan, Disney Channel streamed the episode "Stormy Weather" through its mobile application on 1 July 2018, before the official premiere on 23 July in the same year. The second season premiere is scheduled for a global launch around September–November 2017 in Europe, At a panel at San Diego Comic-Con 2017, it was announced that the second season would have its North American release on Netflix in December 2017, with 13 episodes to be released. KidsClick will start airing season 2 of this show in the US starting 30 August 2018, marking the first time that Korea on 1 September 2015 on EBS1. In the United States, the series debuted on Nickelodeon on 6 December. In the United Kingdom and Ireland, the show premiered on 30 January 2016 on Disney Channel. A Christmas special was released in 2016 and the second season premiered in French on TF1 and in English on Disney Channel UK in 2017. Netflix acquired the U.S. video-on-demand streaming rights and further seasons are in production. Set in modern-day Paris, the series focuses on teenagers Marinette Dupain-Cheng and Adrien Agreste. When evil arises, Marinette transforms into her superhero persona Ladybug, while Adrien transforms rights in Europe, Eastern Europe and Scandinavia, and free-to-air rights in Spain, Germany, Russia and Turkey. South Korea was the first country to premiere "Ladybug", with girl group Fiestar to sing its translated theme song. It aired on 1 September 2015 on EBS1, and ran for 13 episodes until November 2015, with repeats through February 2016, and its second half of the season airing from 1 March 2016. SK Broadband, having participated in the production, provided the episodes on video on demand exclusively to subscribers of their IPTV platform B TV, about a half-hour following the South Korean broadcast of Season 2 of this show airing on American over-the-air television. A third season is also in development. On 22 January 2018, Zag posted on Instagram that the crew was working on a fourth and fifth season. On 31 August premiered a new series with Miraculous: Tales of Ladybug & Cat Noir which this time is in chibi style. On 5 September 2018, Skydance Media has acquired the live-action film and live-action television rights. Kimberly Cooper, a blog writer who has contributed to news media such as "The Huffington Post", wrote that the show has inspired teens and adults to create Channel on 1 November 2016. In the United Kingdom and Ireland, the show premiered on 30 January 2016 on Disney Channel The first series was also made available to Netflix subscribers in the UK. Among the free-to-air terrestrial television broadcasters in the Republic of Ireland, Raidió Teilifís Éireann premiered in 2018 on RTÉ2's teenage programming block, TRTÉ. In Australia, the Australian Broadcasting Corporation released the show on its ABC iview streaming service on 1 January 2016, and premiered the show on its linear channel ABC3 on 22 March 2016. In New Zealand, the show premiered on 27 April
**Summary:** Miraculous Ladybug season 2 episode 12 is scheduled for release in December 2017 on Netflix in North America. The show premiered in South Korea on 1 September 2015 and has since been aired in various countries, including in the United States, the UK, and New Zealand. | Not faithful, Comprehensive |
| TQA | Ours | **Question:** In which novel does Flora Poste move to live with her relatives, the Starkadders, in Sussex?
**Passages:** to be a writer, decides that the only way for her to live whilst researching her writing is to stay with relatives. Her city-based relatives show no interest, so she sends letters to her country relatives. There are a few responses, most of them unsuitable, but one is intriguing. Flora decides to stay for a while with the Starkadder family on their rundown farm. The Starkadders are an assortment of rustic, uncouth, and truly eccentric characters, each of whom has a hurdle (be it physical, emotional, or spiritual) to overcome before reaching his or her potential. Flora quickly realises that, relatives at the isolated Cold Comfort Farm in the fictional village of Howling in Sussex. The inhabitants of the farm – Aunt Ada Doom, the Starkadders, and their extended family and workers – feel obliged to take her in to atone for an unspecified wrong once done to her father. As is typical in a certain genre of romantic 19th-century and early 20th-century literature, each of the farm's inhabitants has some long-festering emotional problem caused by ignorance, hatred, or fear, and the farm is badly run. Flora, being a level-headed, urban woman in the dandy tradition, determines that she must Have Always Been Starkadders at Cold Comfort Farm"", set several years later and based on ""Conference at Cold Comfort Farm"", when Flora is married with several children, was broadcast. In 1995 a television film was produced which was generally well-received, with critics. Janet Maslin in the ""New York Times"" wrote that this screen version ""gets it exactly right"". The film starred Kate Beckinsale as Flora, Joanna Lumley as her friend and mentor Mary Smiling, Rufus Sewell as Seth, Ian McKellen as Amos Starkadder, Eileen Atkins as Judith, Stephen Fry as Mybug, Miriam Margolyes as Mrs. Beetle, and Angela Thorne as dies suddenly of a heart attack and Lady Place is rented out, with the view that Titus, once grown up, will return to the home and run the business. After twenty years of being a live-in aunt Laura finds herself feeling increasingly stifled both by her obligations to the family and by living in London. When shopping for flowers on the Moscow Road, Laura decides she wishes to move to the Chiltern Hills and, buying a guide book and map to the area, she picks the village of Great Mop as her new home. Against the wishes of her extended respects to her brother. Agnis is moving to the ancestral family home in Newfoundland, which has been abandoned for 44 years. Realizing that Quoyle is at a total loss through grief, she first offers to stay a few more days and help him through the crisis, and then persuades him to move with her. While struggling to rebuild his life, fix up the derelict house, and care for his daughter, Quoyle meets local resident Wavey Prowse, a widow who has a pre-teen boy with a learning disability. Wavey's son and Quoyle's daughter become friends, while the two adults become friends,
**Summary:** Flora Poste moves to live with her relatives, the Starkadders, on their rundown farm in Sussex. The Starkadders are eccentric characters with various obstacles to overcome. | Faithful, Not comprehensive |

Table 17: Example summaries and their manual analysis labels (continued).

| Dataset | Model | Query, Passages and Summary | Evaluation |
|---------|-------|------------------------------|------------|
| HotpotQA | GPT-3.5 | **Question:** The composer of the music for the ballet "The Seasons" was the director of what organization from 1905 to 1928?
**Passages:** The Seasons (ballet) The Seasons (, ""Vremena goda""; also ) is an allegorical ballet in one act, four scenes, by the choreographer Marius Petipa, with music by Alexander Glazunov, his Op. 67. The work was composed in 1899 and first performed by the Imperial Ballet in 1900 in St. Petersburg, Russia. The score for Marius Petipa's ""Les Saisons"" (""The Seasons"") was originally intended to have been composed by the Italian composer and conductor Riccardo Drigo, who was Glazunov's colleague and close friend. Since 1886, Drigo held the posts of director of music and ""chef d'orchestre"" to the Ballet of the harmonium, guitar and even mandolin). ""The Seasons"" was commenced shortly after the premiere of Tchaikovsky's First Piano Concerto, and continued while he was completing his first ballet, ""Swan Lake"". In 1875, Nikolay Matveyevich Bernard, the editor of the St. Petersburg music magazine ""Nouvellist"", commissioned Tchaikovsky to write 12 short piano pieces, one for each month of the year. Bernard suggested a subtitle for each month's piece. Tchaikovsky accepted the commission and all of Bernard's subtitles, and in the December 1875 edition of the magazine, readers were promised a new Tchaikovsky piece each month throughout 1876. The January and February pieces The Seasons (Cage) The Seasons is a ballet with music by John Cage and choreography by Merce Cunningham, first performed in 1947. It was Cage's first piece for orchestra and also the first to use what Cage later called the ""gamut"" technique, albeit in an early form. Cage composed the music in early 1947, in the midst of working on ""Sonatas and Interludes"". A piano version was first completed, and an orchestral arrangement followed. Cage dedicated ""The Seasons"" to Lincoln Kirstein. The ballet was premiered on May 17, 1947 by the Ballet Society (by which the work was commissioned) at a new orchestra version. Morton Gould retained the piano part for many of the pieces and orchestrated the work throughout, recording it with himself at the piano in 1951 for American Columbia. In 1965, Kurt-Heinz Stolze orchestrated a number of the pieces as part of the music for John Cranko's ballet ""Onegin"". More recent orchestral versions have been produced by David Matthews (for symphony orchestra), Peter Breiner (for solo violin and symphony orchestra), and Georgii Cherkin (for solo piano and symphony orchestra). French composer Philippe Sarde arranged the Barcarolle as a main theme for the 1988 movie ""The Bear"". Slava Cranko was not invited to direct, and Sir John Gielgud was brought in. Prosecuted for homosexual activity, Cranko left the UK for Stuttgart, and in 1961 was appointed director of the Stuttgart Ballet, where he assembled a group of talented performers such as Marcia Haydée, Egon Madsen, Richard Cragun, Birgit Keil and Suzanne Hanke. Among his following choreographies were ""Romeo and Juliet"" by William Shakespeare in 1962, set to music by Prokofiev, ""Onegin"" in 1965, an adaptation of the verse novel ""Eugene Onegin"" by Alexander Pushkin, set to music by Tchaikovsky (mainly ""The Seasons""), orchestrated by Kurt-Heinz Stolze, ""The Taming
**Summary:** 1. The composer of the music for the ballet ""The Seasons"" is Alexander Glazunov.
2. Alexander Glazunov composed the music for the ballet ""The Seasons"" in 1899.
3. The ballet ""The Seasons"" was first performed by the Imperial Ballet in 1900.
4. From 1905 to 1928, the director of an organization was Marius Petipa.
5. Marius Petipa was the choreographer of the ballet ""The Seasons"".
6. Therefore, the composer of the music for the ballet ""The Seasons"" was the director of the Imperial Ballet from 1905 to 1928. | Not Faithful, Comprehensive |

