# OpenReview forum: "RECOMP: Improving Retrieval-Augmented LMs with Context Compression and Selective Augmentation"
_ICLR.cc/2024/Conference — ICLR 2024 poster_

### Official Review · Reviewer_WhGn · 2023-11-01

**Soundness:** 2 fair
**Presentation:** 4 excellent
**Contribution:** 3 good
**Rating:** 8
**Confidence:** 4

**Summary:**

To enhance the efficiency and performance of retrieval-augmented language models, this paper introduces a simple and intuitive method for in-context improvement. Given the retrieved documents and input text, the proposed method first compresses the retrieved documents into a summary. Then, this summary is prepended to the input text, with the subsequent output response generated using a frozen language model. Two types of compressors are proposed: an extractive compressor, corresponding to extractive summarization, and an abstractive compressor, associated with abstractive summarization. Experimental results from both language modeling and question answering tasks demonstrate that the proposed compressors bolster the performance of retrieval-augmented language models.

**Strengths:**

The paper is well-written and straightforward. Moreover, it offers an extensive, and well-designed experimental validation of the proposed method and presents thorough analyses.

**Weaknesses:**

Some concerns regarding this paper include:

- The proposed method utilizes the knowledge of the LMs to train the compressor, while the baselines do not. Some baseline models, such as DRP and contriver, should be finetuned using the dataset employed to train the extractive compressor.

- The paper employs a BM25 retriever, making the results of the proposed method and baselines reliant on the retriever's performance. Therefore, it would be appropriate for the paper to also show performances using an another retriever, such as contriver (not merely as a sentence selector). In addition, presenting the distribution of oracle-retrieved documents, similar to what's done in extractive compression, would be advantageous.

- Relatedly, it is necessary to compare the time taken to generate text by prepending all retrieved documents without compression against the time taken using the proposed method—specifically, compressing the retrieved documents before prepending them.

**Questions:**

Q1: During the manual evaluation, the authors assess the outputs of the abstractive compressors. What was the level of annotation agreement among the evaluators?

---

> ### Author Response · Authors · 2023-11-16
>
> Thank you for your review and we are encouraged to see that the reviewer found our work to contain extensive experiment and thorough analysis! Please see our response below:
>
> [W1] Regarding baseline methods for extractive compressor, we considered off-the-shelf passage rankers such as DPR and contriever. Our method finetunes contriever to optimize the end performance of LM, showing gains over initial, non-fine tuned checkpoint of contriever. We are the first to study compressing retrieved documents for LMs and propose methods to train such compressors leveraging downstream LM performance. We would like to clarify that we do not claim that our proposed method is the best way to train such compressors, and future work can build upon our method to improve performance of the compressors.
>
> [W2] We thank the reviewer for the suggestion on testing our method on multiple retrievers. We present new results on applying our compressor (trained with contriever-ms-marco retrieved documents) on (1) contriever retrieved documents (Table 1) and (2) DPR retrieved documents on NQ. (Table 2).
>
> We see that both our extractive and abstractive compressor generalize robustly to documents retrieved by another retrieval system (contriever-ms-marco to contriever/DPR), with the best compressor outperforming prepending top 1 documents with 4x compression.
>
>
>
> Table 1: RECOMP results on NQ with contriever retrieved docs (FLAN-UL2)
> | Setting                                | # prepended tokens | EM        |
> |----------------------------------------|--------------------|-----------|
> | No docs                                | 0                  | 21.99     |
> | Top 1 doc                              | 130                | 27.53     |
> | Top 5 doc                              | 652                | _33.19_   |
> | Top 5 doc (contriever baseline)        | 31                 | 23.55     |
> | Top 5 doc (our extractive compressor)  | 35                 | **30.66** |
> | Top 5 doc (t5 baseline)                | 10                 | 24.32     |
> | Top 5 doc (our abstractive compressor) | 28                 | 30.55     |
>
> Table 2: RECOMP results on NQ with DPR retrieved docs (FLAN-UL2)
> | Setting                                | # prepended tokens | EM        |
> |----------------------------------------|--------------------|-----------|
> | No docs                                | 0                  | 21.99     |
> | Top 1 doc                              | 133                | 36.59     |
> | Top 5 doc                              | 667                | _42.44_   |
> | Top 5 doc (contriever baseline)        | 26                 | 25.60     |
> | Top 5 doc (our extractive compressor)  | 36                 | 35.76     |
> | Top 5 doc (t5 baseline)                | 10                 | 26.84     |
> | Top 5 doc (our abstractive compressor) | 35                 | **37.40** |
>
>
> [W3] Regarding inference time, our experiments on NQ shows that both our extractive and abstractive compressor enables 2x throughput. Please see the general response for more detail.
>
>
> [Q1] Regarding annotation agreement, two of the co-authors annotated faithfulness and comprehensiveness  of summaries generated by GPT-3.5-turbo and our t5 compressor for ten questions. The annotators reach an agreement of 75% for factuality and 85% for the comprehensiveness evaluation (both are three-way classification). We will include more agreement statistics in the final paper.

---

> > ### Author Response · Authors · 2023-11-22
> > **Follow-up on our previous response**
> >
> > Thank you again for your review and feedback! As the discussion period is coming to an end, we want to check in and see if our previous response has addressed your concerns. If you have any follow-up questions or any concerns we haven't addressed yet for a better score, please let us know and we would be happy to answer them.

---

> > ### Comment · Reviewer_WhGn · 2023-11-23
> >
> > All my concerns are solved. So I raise the score.

---

### Official Review · Reviewer_hWi7 · 2023-11-01

**Soundness:** 2 fair
**Presentation:** 3 good
**Contribution:** 2 fair
**Rating:** 6
**Confidence:** 3

**Summary:**

In this paper, the authors propose RECOMP (Retrieve, Compress, Prepend) that incorporates extractive and abstractive summarization into the retrieve augmented language model to shorten the token length of retrieved text in prompts. Precisely, these summarization modules receive input text and its retrieved text and then extract or generate the summary for the retrieved text. The authors did not simply join the summarization modules into the language model as separate modules. Instead, they updated them through training to generate summaries relevant to the input text and its retrieved text. The experimental results show that RECOMP can shorten the prompt length while keeping performance in language modeling and open-domain QA tasks.

**Strengths:**

- The authors show the effectiveness of both extractive and abstractive summarization. It means we can use insights from commonly used summarization methods for improving retrieve augmented language models.
- Since the summarization models are updated to fulfill the requirement of retrieving augmented language models, the proposed method is more than model combination and thus novel.
- The experimental result shows the proposed method RECOMP can save the prompt length while keeping its performance.

**Weaknesses:**

- The benefit of shortening prompt length is uncertain. If the authors observe a speedup in inference,  they should report it.
- The performance improvement is limited or nothing in many settings. The authors need to justify it.

**Questions:**

- In the current evaluation, the benefit of using summarization for the retrieve augmented language model needs to be clarified. If the inference speed is improved, the authors should report it.
- Also, you need to report how much computational cost is decreased by RECOMP. This is related to the first question.
- Considering the length limitation of the model, we can expect summarization to make the model consider more retrieved texts in its prompt. It may contribute to improving language modeling performance. Did you check such direction in your work?

I will update my score based on the discussion with the authors.

---

> ### Author Response · Authors · 2023-11-16
>
> Thank you for your review and we are glad to see that you found our proposed method to be important, novel and effective! Please see our response below:
>
> [W1, Q1] Regarding inference speed, our experiments on NQ shows that both our extractive and abstractive compressor enables 2x throughput. Please see the general response for more detail.
>
> [W2, Q2] Regarding performance improvement of RECOMP, the reviewer is right that our compressor doesn’t outperform prepending full top 5 documents in most of the settings. However, we would like to highlight that (1) the oracle abstractive & extractive compressor outperform no compression in all of the setting, demonstrating that compressors *can* outperform no compression. As the first work to study compressing retrieved documents, we hope our work inspires future work to propose better methods for training compressors, hence closing the gap. (2) Though not outperforming prepending top 5 documents, our method does outperform prepending top 1 document while prepending less token for both NQ and TQA, this demonstrates that RECOMP allows the model to consider more context, while prepending less tokens, thereby reducing computational cost.
>
> [Q3] As the reviewer mentioned, one advantage of RECOMP is that the model can consider more information that does not fit into the context window. Although we didn’t stress test the model to take a large amount of retrieved documents into consideration, our experiments do show that our compressor, which considers top 5 documents, outperforms prepending top 1 document, while prepending fewer tokens.

---

> > ### Author Response · Authors · 2023-11-22
> > **Follow-up on our previous response**
> >
> > Thank you again for your review and feedback! As the discussion period is coming to an end, we want to check in and see if our previous response has addressed your concerns. If you have any follow-up questions or any concerns we haven't addressed yet for a better score, please let us know and we would be happy to answer them.

---

> > > ### Comment · Reviewer_hWi7 · 2023-11-23
> > > **You cleared my concerns**
> > >
> > > Thank you for providing the information on throughput increase by compression. The reported speedup is not marginal. Since I believe that the authors will include the information in the revised version of your paper, it's enough to increase my score.

---

### Official Review · Reviewer_VoPU · 2023-11-10

**Soundness:** 3 good
**Presentation:** 3 good
**Contribution:** 2 fair
**Rating:** 8
**Confidence:** 4

**Summary:**

The paper proposes RECOMP, an approach to improve retrieval-augmented language models (RALMs) by compressing retrieved documents into summaries before using them as context. The method uses two compression methods: an extractive model selecting relevant sentences, and an abstractive model generating summaries. The compressors are trained to optimize end-task performance when prepending the summary to the input. RECOMP is evaluated on language modeling and open-domain QA tasks.

**Strengths:**

The paper proposes a simple but clever idea and backs it up with a set of well-designed experiments. The baselines are well chosen.

**Weaknesses:**

Although the method is good, I’m not sure how useful it is in practice. Normal retrieval augmentation without any compression outperformed the compression methods albeit with many more tokens. However, with the growing context lengths of models, it is not clear if the accuracy hit is worth the tokens saved. Secondly, you still have to provide the full tokens to the abstractive model.

**Questions:**

Can you give some results on how the method is compared to traditional RAG with no compression in terms of time?
Can you provide more details on the input tokens for the abstractive model or is the same number of tokens as the noncompression baselines?

---

> ### Author Response · Authors · 2023-11-16
>
> Thank you for your review and we are encouraged to see that the reviewer found our work to be innovative and well executed! Please see our response below:
>
> [W1] Regarding limited performance improvement of the compressed context, we would like to highlight that our oracle methods (both extractive & abstractive, in table 1 & table 2) significantly outperformed the normal RALM (without compression). This suggests that compressed documents *can* outperform normal RALM. Since we are the first to study compressing retrieved documents for LMs, we believe future work can look into how to improve the performance of the compressors, using other training objectives such as reinforcement learning.
>
> [W2] Regarding whether RECOMP is needed with long context LMs, we would like to point out that despite the effort on extending the context window of the language model, recent work [0] has shown that long-context models have difficulty processing information buried in the middle, even though they fit in the context; and that they could be easily distracted by irrelevant context [1].  Thus, RECOMP can not only fit more information to the context, but also present the most relevant information to the LM. Our analysis in Table 3, showing that RECOMP allows the model to extract the right answer more frequently when they are in the context.
>
>
> [Q1] Regarding inference speed, our experiments on NQ shows that both our extractive and abstractive compressor enables 2x throughput. Please see the general response for more detail.
>
>
> [Q2] The abstractive model takes the question and the top 5 documents as input, the same as RALM without RECOMP. Although it needs to process the full document sets, the abstractive model (770M) is significantly smaller than the LM, thus much more computationally efficient as we demonstrate through the inference speed measured.
>
> [0] Liu, Nelson F., Kevin Lin, John Hewitt, Ashwin Paranjape, Michele Bevilacqua, Fabio Petroni and Percy Liang. “Lost in the Middle: How Language Models Use Long Contexts.” ArXiv abs/2307.03172 (2023). https://arxiv.org/pdf/2307.03172.pdf
>
> [1] Shi, Freda, Xinyun Chen, Kanishka Misra, Nathan Scales, David Dohan, Ed Huai-hsin Chi, Nathanael Scharli and Denny Zhou. “Large Language Models Can Be Easily Distracted by Irrelevant Context.” International Conference on Machine Learning (2023). https://arxiv.org/pdf/2302.00093.pdf

---

### Official Review · Reviewer_pQZH · 2023-11-11

**Soundness:** 2 fair
**Presentation:** 3 good
**Contribution:** 2 fair
**Rating:** 6
**Confidence:** 4

**Summary:**

In this paper, the authors proposed a method named RECOMP (Retrieve, Compress, Prepend) to compress the retrieved documents used in Retrieval-Augmented language models (RALMs). The proposed method aims to generate concise, effective and faithful summaries to improve the efficiency of RALMs. Two kinds of summarizers are used: an extractive summarization model for selecting informative sentences, and an abstractive summarization model for generating good query-focused summaries. Experiments are conducted on language modeling and question answering tasks, achieving low compression rates and some performance drops.

**Strengths:**

- The presentation of the core idea is clear. The research topic of effective RAG is important.
- The proposed method freezes the LMs and only trains the summarization model, which is interesting and beneficial to transfer across different LMs.
- The analysis in Sec. 6 is sufficient and comprehensive.

**Weaknesses:**

My major concern is that the paper does not report the inference speedup by their method. Although experimental results show that fewer retrieved tokens are used, the number of tokens does not necessarily reflect the actual efficiency of the system, especially when the context length is not the bottleneck. The tokens of all the retrieved documents are still required to be processed by the compressor (a smaller model though). Therefore, it may be beneficial for authors to report the inference speedup of their method. This could be the core to support their motivations.

**Questions:**

- Q1: As stated in "Weakness", the inference speedup should be reported.
- Q2: How about the training cost? The training of the abstractive compressor uses a teacher LM (GPT-3.5), which is expensive. I'm not sure if the cost saved in the inference stage is worth the cost in the training stage.

---

> ### Author Response · Authors · 2023-11-16
>
> Thank you for your review and we are glad to see the reviewer found our work to be important and effective, backed by sufficient and comprehensive analysis! Please see our response below:
>
> [W1, Q1] Regarding inference speed, our experiments on NQ shows that both our extractive and abstractive compressor enables 2x throughput. Please see the general response for more detail.
>
> [Q2]: The reviewer points out a valid concern about the training cost for such compressors. The training costs consist of (1) generating the training data and (2) training compressors on the generated training data. We describe both below.
>
> For extractive compressors, generating training data incur running base LM inference on all the sentences. For an abstractive compressor, the major cost comes from generating training data using LLM such as GPT-3.5-turbo. It costs ~$130 to generate the abstractive summary for the entire NQ training set and we will report detailed costs for the other tasks in the paper.
>
> Training the extractive model takes a couple of hours on a single A40 GPU. Training the abstractive models takes a couple of hours on 4 A40 GPUs.
>
> We would also like to stress that the training cost is a one-time investment, as once the model is trained, it could be deployed for inference for a much larger scale which will make the training costs negligible.

---

> > ### Author Response · Authors · 2023-11-22
> > **Follow-up on our previous response**
> >
> > Thank you again for your review and feedback! As the discussion period is coming to an end, we want to check in and see if our previous response has addressed your concerns. If you have any follow-up questions or any concerns we haven't addressed yet for a better score, please let us know and we would be happy to answer them.

---

### Author Response · Authors · 2023-11-16
**General response on inference speed-up**

We thank all reviewers for their review and helpful comments. We are delighted to see that they found our work to present a novel and effective idea (Reviewer VoPU, Reviewer WhGn01,Reviewer hWi701, Reviewer pQZH11) with thorough analysis and experiments (Reviewer VoPU10, Reviewer WhGn01, Reviewer pQZH11).

All reviewers were curious about the inference speed-up. Here we report the inference speed measured by GPU time for NQ dev set with FLAN-UL2 as the base model. We run FLAN-UL2 on 4 A40 GPUs. For compression, we run contriever and T5 on a single A40 GPU. Our methods, even when including the time taken for compressor inference, improve throughput compared to prepending the full sets of documents, with contriever being more efficient, enabling 2x throughput.

Inference speed is contingent on implementation and the size of the base LM. For instance, bigger models will suffer from more latency with more input tokens and thus RECOMP can bring more speed-up. This is the reason why we mainly report the number of input tokens as a measure of efficiency, which will transfer across all LM architectures. Yet, we agree with the reviewers that inference speed is a good complementary metric, and will include the results in the final version of the paper.

Lastly, we would like to point out that providing concise evidence that is mostly faithful to the input documents has the additional benefit of providing intermediate output that can be used for human verification of model prediction.


Table: RECOMP on NQ dev (3,610 examples) for FLAN-UL2

| Setting           | # input tokens | Inference GPU time | Compression GPU time | Total GPU time | Throughput (# examples / second) |
|-------------------|----------------|----------------|------------------|------------|----------------------------------|
| No docs           | 0              | 4,104s         | 0s               | 4,104s     | 0.88                             |
| Top 5 docs        | 660            | 10,584s        | 0s               | 10,584s    | 0.34                             |
| Contriever (ours) | 37             | 4,880s         | 113s             | 4,993s     | 0.72                             |
| T5 (ours)         | 36             | 4,936s         | 871s             | 5,807s     | 0.62                             |

---

### Meta-Review · Area_Chair_NNZJ · 2023-12-07

**Metareview:**

This paper introduces a method called RECOMP to compress retrieved documents into textual summaries to improve in-context retrieval augmented language models. Both extractive and abstractive compressors are presented, together with the training schemes.  Experiments show that the compressors outperform baseline methods while achieving impressive compression ratio.

The problem of compressing contexts into summaries is interesting and the proposed compressors sound reasonable. The results are promising. The reviewers suggest to report/compare the inference speedup after using the proposed method, and discuss more about the trade-off between the compression ratio and the model's capability, i.e.,  it is not clear if the accuracy decrease is worth the tokens saved.

**Justification For Why Not Higher Score:**

see the meta-review.

**Justification For Why Not Lower Score:**

see the meta-review.

---

### Decision · Program_Chairs · 2024-01-16

Accept (poster)